# Deciphering Bacterial Community Structure, Functional Prediction and Food Safety Assessment in Fermented Fruits Using Next-Generation 16S rRNA Amplicon Sequencing

**DOI:** 10.3390/microorganisms9081574

**Published:** 2021-07-23

**Authors:** Bashir Hussain, Jung-Sheng Chen, Bing-Mu Hsu, I-Tseng Chu, Suprokash Koner, Tsung-Hsien Chen, Jagat Rathod, Michael W. Y. Chan

**Affiliations:** 1Department of Biomedical Sciences, National Chung Cheng University, Chiayi 621, Taiwan; bashir.aku@gmail.com (B.H.); suprokashkoner22@gmail.com (S.K.); biowyc@ccu.edu.tw (M.W.Y.C.); 2Department of Earth and Environmental Sciences, National Chung Cheng University, Chiayi 621, Taiwan; 3Department of Medical Research, E-Da Hospital, Kaohsiung 824, Taiwan; nicky071214@gmail.com; 4Department of Internal Medicine, Division of Cardiology, Ditmanson Medical Foundation Chia-Yi Christian Hospital, Chiayi 621, Taiwan; 02500@cych.org.tw; 5Department of Internal Medicine, Ditmanson Medical Foundation Chia-Yi Christian Hospital, Chiayi 621, Taiwan; cych13794@gmail.com; 6Department of Earth Sciences, National Cheng Kung University, 1st University Road, Tainan 701, Taiwan; jagat2006@gmail.com

**Keywords:** fermented fruits, bacterial diversity, food safety, functional prediction, next-generation sequencing, opportunistic pathogens, 16S rRNA metagenomics

## Abstract

Fermented fruits and vegetables play an important role in safeguarding food security world-wide. Recently, robust sequencing-based microbial community analysis platforms have improved microbial safety assessment. This study aimed to examine the composition of bacteria and evaluate the bacterial safety of fermented fruit products using high-throughput 16S-rRNA metagenomic analysis. The operational taxonomic unit-based taxonomic classification of DNA sequences revealed 53 bacterial genera. However, the amplicon sequencing variant (ASV)-based clustering revealed 43 classifiable bacterial genera. Taxonomic classifications revealed that the abundance of *Sphingomonas*, which was the predominant genus in the majority of tested samples, was more than 85–90% among the total identified bacterial community in most samples. Among these identified genera, 13 low abundance genera were potential opportunistic pathogens, including *Acinetobacter, Bacillus, Staphylococcus, Clostridium, Klebsiella, Mycobacterium, Ochrobactrum, Chryseobacterium, Stenotrophomonas*, and *Streptococcus*. Of these 13 genera, 13 major opportunistic pathogenic species were validated using polymerase chain reaction. The pathogens were not detected in the samples of different stages and the final products of fermentation, except in one sample from the first stage of fermentation in which *S. aureus* was detected. This finding was consistent with that of ASV-based taxonomic classification according to which *S. aureus* was detected only in the sample from the first stage of fermentation. However, *S. aureus* was not significantly correlated with the human disease pathways. These results indicated that fermentation is a reliable and safe process as pathogenic bacteria were not detected in the fermentation products. The hybrid method reported in this study can be used simultaneously to evaluate the bacterial diversity, their functional predictions and safety assessment of novel fermentation products. Additionally, this hybrid method does not involve the random detection of pathogens, which can markedly decrease the time of detection and food safety verification. Furthermore, this hybrid method can be used for the quality control of products and the identification of external contamination.

## 1. Introduction

Fermentation is a cost-effective method to preserve perishable and seasonal fruits, which play a vital role in safeguarding food security world-wide [1,2]. Fermented products, which have several advantages, including increased nutritional value, are a major source of minerals, vitamins, and carbohydrates to the vulnerable population [3,4]. Moreover, fermented products of fruits and vegetables are an integral part of most cuisines worldwide [5]. Various types of fermented foods are estimated to comprise up to 30% of the food supply [6]. However, the microbiological safety of these products must be ensured before public consumption [7]. Generally, fermented products are considered microbiologically safe as the harsh fermentation conditions, such as low pH, high temperature, and increased sugar and salt contents prevent the growth of pathogenic microbes [6,8]. However, several studies have demonstrated that some pathogens, such as *Listeria monocytogenes, Clostridium* spp., *Salmonella* spp., *Staphylococcus* spp., *Bacillus* spp., and *Escherichia coli* survive and escape during fermentation and remain infectious [9,10,11,12]. The fermentation conditions, such as the time-temperature profile, water activity, and pH determine the microbial profile of the fermented products [13,14]. Additionally, other factors, including poor sanitation, starter culture quality, and raw material quality, contribute to the microbial profiles of fermented food [14,15].

Recently, there is increased consumer awareness on food safety and health benefits, including food quality and microbial traceability and safety [16,17]. The recent advances in next-generation sequencing (NGS) have revolutionized the traceability and characterization of a broad range of pathogens in various fields, including epidemiology, diagnostics, food quality, and forensics [18,19]. The fermentation process involves nonfunctional microbial communities, active microbial communities, and abiotic factors, which decrease the emergence of pathogenic microbes during the fermentation processes and the final products [14,19]. Some bacteria, especially the members of *Enterococcus* and *Enterobacteriaceae*, produce toxic metabolites, such as biogenic amines during the fermentation process [19]. Thus, the consumption of food products containing toxic metabolites and viable pathogens may lead to various foodborne illnesses [19]. Some fermented food-associated microorganisms, such as *Enterococcus*, *Staphylococcus*, and *Lactobacillaceae* can harbor transmissible antibiotic resistance genes [20] Hence, there is a need to characterize the function and diversity of microbes in fermented food products to improve the quality and safety measures of the final product [21]. Traditional approaches, such as biochemical and physiological methods used for routine enumeration and characterization of pathogens are expensive, laborious, and time-consuming [22,23]. Additionally, traditional methods cannot provide an accurate pathogen risk assessment as they cannot capture the complete microbial diversity [24]. Molecular approaches based on the DNA/RNA signatures of food matrices enable the understanding of microbial complexity and the ecology of the fermentation process at the molecular level. Additionally, molecular approaches can capture the complete microbial population, including the viable but non-culturable microbes [25,26]. Furthermore, molecular approaches can detect injured or stressed microbial populations [27]. The advances in NGS-based technologies have decreased the cost of microbial analysis and increased the resolution of microbial community analysis [28,29]. Recent studies have utilized NGS to successfully examine the microbial ecology of various fermentation processes of food items, such as tea, milk, wine, fruits, and vegetables [21,24,30,31]. However, most studies have focused on the characterization of functional and beneficial microbial diversity. There are limited studies examining the microbiological safety of fermentation products using NGS techniques [32].

This study aimed to investigate the total bacterial community and their functional prediction along with the identification of potential pathogens in fermented fruits at different stages of fermentation. In this study, a metagenomics approach was used to target the V3 and V4 regions of the 16S rRNA gene to determine the change in bacterial diversity in the different stages of fermentation and the final fermentation products. The identification of unique bacterial communities, including pathogens, and their predicted functions can aid in improving food safety. This study characterized the bacterial community structures and their safety parameters, which can increase the consumer acceptance of fermented fruit-based products.

## 2. Materials and Methods

### 2.1. Samples and Chemical Analysis

In this study, sampling was performed during various stages of the fruit fermentation process using the intermediate and finished products from a commercial fermented beverage production company (Taiwan Enzyme Village Co., Ltd. Chiayi, Taiwan) located in Chiayi County, Taiwan. The fruit fermentation of pineapple, orange, and banana is carried out into two stages, which ultimately lead to the final product. In the 1st stage of fermentation, these fruits are washed and cut in fermentation tanks followed by the addition of sugar, yeast, lactic acid bacteria (LAB), and acetic acid bacteria (AAB), then followed by mixing vigorously for 24 h. The concentration of yeast, LAB, and AAB additives was 0.0025%, whereas, the concentration of sugar was 23.2% in a total of 1 metric ton of fermented fruit. However, pineapple samples without starter culture did not contain yeast, LAB, and AAB in the fermentation tanks. Finally, these fermentation tanks were kept at room temperature under anaerobic conditions to accomplish the 1st stage of the fermentation process. The total fermentation duration of pineapple, banana and orange in the 1st stage was approximately 180, 150, and 365 days, respectively. The sampling time was based on 1/3 and 2/3 of the fermentation duration for pineapple and banana, whereas orange was collected at 1/2 and 3/4 of the fermentation duration. The 2nd stage of fermentation was performed by mixing the above fermented products from the 1st stage in a fermentation tank and kept for approximately one year at room temperature under anaerobic conditions. Real-time temperature detection was used to ensure the range of room temperature (in between 24–28 °C) during fermentation. The final product of fermentation was a mixture of liquid products from the 2nd stage of fermentation. In total, 28 samples (40 mL of fermented broth for each sample) were aseptically collected from their respective fermentation tanks and final product into sterilized containers and transported to the laboratory under controlled temperature. Description of the samples, fermentation processes and sampling time are provided in Table 1. Additionally, a brief description of chemical analysis from the 2nd stage of fermentation and the final product are shown in Appendix A.

### 2.2. Genomic DNA (gDNA) Extraction and Bacterial 16S rRNA Amplicon Analysis

The fermented broth (2 mL) was thoroughly mixed, aseptically transferred to a sterilized tube, and centrifuged at 12,000× *g* for 3 min. The supernatant was discarded and the bacterial cells were lysed to extract gDNA using a commercial auto DNA extractor kit (MagPurix Bacterial DNA Extraction Kit, ZP02006, Taipei, Taiwan), following the manufacturer’s instructions. The purity and concentration of the extracted gDNA were determined using a Nanodrop 2000 spectrophotometer (Thermo Fisher Scientific Inc., Wilmington, DE, USA) at 230–280 nm. The quality of gDNA was examined using gel electrophoresis (1.5% gel in Tris-acetate ethylenediaminetetraacetic acid buffer) at 110 V for 30 min. DNA bands were visualized under ultraviolet light. The purified gDNA was stored at −20 °C for further analysis.

### 2.3. Sequencing, Library Construction, and 16S rRNA Amplicon Data Analysis

The libraries for the amplified V3–V4 regions of 16S rRNA hypervariable regions were constructed using the 341F and 805R primer sets (with some modifications) with the Illumina adapter overhang sequence attached at the 5′ end of the primers. The sequences of forward primers used in this experiment were as follows: (16S_341F) 5′-TCGTCGGCAGCGTCAGATGTGTATAAGAGACAGCCTACGGGNGGCWG CAG-3′, (16S_341F _N) 5′-TCGTCGGCAGCGTCAGATGTGTATAAGAGAC AGnCCTACGGGNGGCWGCAG-3′, (16S_341F _N N) 5′-TCGTCGGCAGCGTCAG ATGTGTATAAGAGACAGnnCCTACGGGNGGC WGCAG-3′, and (16S_341F _NNN) 5′-TCGTCGGCAGCGTCAGATGTGTATAAGAGACAGnnnCCTACGGGNGG CWGCAG-3′. The reverse primer sequences were as follows: (16S_805R) 5′-GTCTCGTGGGCTCGGAGATGTGTATAAGAGACAGGACTACHVGGGTA TCT AATCC-3′, (16S_805R _N_) 5′-GTCTCGTGGGCTCGGAGATGTGTATAAGAGA CAGnGACTACHVGGGTATCTAATCC-3′, (16S_805R _NN) 5′-GTCTCGTGGGC TCGGAGATGTGTATAAGAGACAGnnGACTACHVGGGTATCTAATCC-3′, and (16S_805R _NNN) 5′-GTCTCGTGGGCTCGGAGATGTGTATAAGAGACAG nnnGACTACHVGGGTATCTAATCC-3′. The amplification was performed in triplicates, following the previously reported method [33]. The optimal PCR conditions were as follows: 95 °C for 3 min, followed by 30 cycles of 95 °C for 30 s (denaturation), 55 °C for 30 s (annealing), 72 °C for 30 s (primer extension), and 72 °C for 5 min (elongation). The quantity and quality of amplified DNA were assessed using the standard quality checks mentioned above. Next, the amplicons (20 µL) from each sample were subjected to sequencing using the pair-end method with the MiSeq Illumina platform (Illumina Inc., San Diego, CA, USA), following standard protocol at the National Yang-Ming University Genome Research Center. The DNA libraries were ligated with the sequencing adapters and index using the Nextera XT sample preparation kit (Illumina), following the manufacturer’s instructions. The sequence data containing reverse and forward reads were aligned using CLC bio plate form (Genomic Workbench v.8.5) and the FASTA files were generated as described in our previous study [34]. The FASTA files were further processed using two standard 16S-metagenomics pipelines. The USEARCH system was used to remove chimeric sequences. The sequences were clustered into operational taxonomic units (OTUs) with a 97% similarity threshold against the Green Genes database. The QIIME2 system was used for the sequence quality control and amplicon sequence variant (ASV)-based classification [35]. Finally, the relative abundance of microbes at the genus level in each sample was obtained using the QIIME2 view. Furthermore, the significant difference in the relative abundance at the genus level at each fermentation stage and in the final fermentation product was analyzed using the statistical analysis of taxonomic and functional profiles (STAMP) software [36]. The significant relative abundance at the genus levels was examined using the two-tailed Welch’s *t*-test (*p* < 0.05).

### 2.4. Metagenomic Functional Prediction Based on 16S rRNA Gene Data

To examine the potential metabolic function of microorganisms at different stages of fermentation, the representative sequence and denoised ASV abundance table were used. These tables were input into the phylogenetic investigation of communities’ reconstruction of Unobserved States (PICRUSt2) pipeline (https://github.com/picrust/picrust2/, accessed on 15 November 2020) using KEGG (Kyoto Encyclopedia of Genes and Genomes). All ASVs with nearest-sequenced taxon index (cut-off value > 2) were removed by default for reliable annotation of metabolic functions using the KEGG reference database as previously described [37]. To improve the accuracy and reliability of the KEGG pathways, a previously described web-based tool (MicrobiomAnalyst) was used to remove extremely low abundant and variant KEGG gene orthologs (KOs) (based on <4 KO filtration) from each sample [38]. The results of levels 1 and 2 KEGG pathways were plotted in the R environment using the ggplot2 package. The significant shift in the bacterial community at each fermentation stage and fermentation product was analyzed using two-tail Welch’s t-test with STAMP software (*p* < 0.05) with 95% confidence intervals. Pearson correlation analysis was performed using IBM SPSS Statistics 24 (IBM, Armonk and North Castle, NY, USA) to evaluate the significant correlations between bacterial diversity and potential functional prediction considering *p* ranging 0.01–0.05.

### 2.5. Detection Methods for Suspected Pathogens after 16S-rRNA Gene Surveys

In this study, species identification of suspected pathogens in fermented samples were validated by various specific genes by targeting PCRs. *Acinetobacter baturnannii* was validated using intergenic spacer (ITS) PCR analysis as described previously [39]. *Pseudomonas aeruginosa*, *Stenotrophomonas maltophilia*, *Gardmerella vaginalis*, and *Chryseobacterium indologenes* were validated using 6S-23S ribosomal DNA region PCR analysis as described previously [40,41,42,43]. The specific targeting *nuc, bal, recA, tyrB* and *pneumolysin* genes PCRs were used to validate *Staphylococcus aureus, Bacillus cereus sensu lato s.l, Ochrobactrum anthropic, Klebsiella pneumoniae*, and *Streptococcus pneumoniae,* respectively [44,45,46,47,48]. *Clostridium difficile* was validated by two genes (*tcdA, tcdB*) PCRs [49]. The conditions for the PCR amplification protocols are described in Appendix A.

## 3. Results

### 3.1. Sequence Depth and Bacterial Biodiversity in the Fermented Fruit Samples with Respect to Fermentation Processes Based on 16S rRNA Gene Amplicons

The 16S rRNA amplicon sequencing targeting the V3 and V4 regions was performed to examine the bacterial community diversity and composition in each fermentation stage sample. After quality filtering and the removal of chimeric sequences, a total of 785,339 sequence reads (average 28,047 per sample) ranging from 4595–75,976 were obtained from 28 samples (Appendix A) and the rarefaction of these sequences was performed at the lowest sequence depth (4595) to compare bacterial diversity among the fermentation stages and final products (Appendix A). The average good’s coverage was more than 99% at this sequencing depth, indicating enough resolution to analyze the downstream analysis such as alpha and beta diversity estimations (Appendix A).

The bacterial diversity and richness among the fermentation stages and final product was elevated using various alpha diversity indices as shown in Figure 1. Among the alpha diversity indices, the Observed (Figure 1A), Chao1 (Figure 1B), and ACE (Figure 1C) revealed higher diversity richness associated with the 1st stage of fermentation and a decrease diversity was observed with increasing the fermentation duration. However, none of these indices showed significance diversity richness among the fermentation stages and final products. Additionally, Simpson (Figure 1D) and Shannon (Figure 1E) alpha diversity indices revealed higher diversity abundance associated with the 2nd stage followed by 1st stage compared to final product of fermentation. However, only Simpson alpha diversity index showed significance among the fermentation stages and final products.

The beta diversity analysis based on Bray–Curtis distances revealed high variation among the bacterial diversity associated with the fermentation stages and product as the result showed a distinct clustering pattern (Figure 1F) indicating different bacterial communities among the experimental groups. The PC1 of ordination representing the 1st stage of fermentation showed the highest variation followed by the second stage of fermentation (PC2: 25.7%) as compared to the final product (PC3: 12%), which is inconsistent with the result of alpha diversity. Additionally, PARMANOVA was applied to know the significant analysis among the fermentation stages and final product, which showed significant variation (*p* < 0.05) in the bacterial beta diversity among the fermentation stages and final product.

### 3.2. Bacterial Taxonomic Abundance in the Fermented Fruit Samples Evaluated Based on 16S rRNA Gene Amplicons

Genus level OTU based classification revealed a total of 52 classifiable genera as shown in Figure 2. Among them *Sphingomonas* was the most predominant bacteria, which accounted for more than 80% of the total bacteria, in most fermentation samples. The second most abundant bacterial genus was *Oribacterium* (18%), which was the predominant genus in Ch3_2nd and Ba5_1st samples with maximum relative abundances of >42% and >98%, respectively. The resolution of low abundance genera, which accounted for less than 10% of the total genera, was limited due to the increased relative abundance of *Sphingomonas*. Therefore, this genus was excluded to visualize and interpret the relative abundance of the remaining 51 genera (Figure 2B). Of the 51 genera, 11 were potential opportunistic pathogens (*Acinetobacter*, *Bacillus*, *Staphylococcus*, *Dietzia*, *Clostridium*, *Klebsiella*, *Mycobacterium*, *Ochrobactrum*, *Chryseobacterium*, *Stenotrophomonas*, and *Streptococcus*). Among these pathogenic bacteria, *Acinetobacter* exhibited a high abundance and was detected in all samples. *Bacillus*, which was detected in 99% of samples, was the second most predominant genus. *Stenotrophomonas*, *Streptococcus*, and *Staphylococcus* were detected in more than 95% of samples irrespective of their relative abundance. In contrast, *Clostridium* was detected in almost 67% of samples. *Klebsiella*, *Chryseobacterium*, *Ochrobactrum*, *Mycobacterium*, and *Deutzia* were detected in more than 10% of samples. Among the 28 samples, Ba1_1st and Ba3_1st exhibited a high distribution of bacterial genera with 15 genera detected in each sample.

To compare the bacterial diversity in each fermentation sample, the Qiime2 pipeline was used. After quality filtering, 193,323 sequence reads were obtained, which were classified into 43 identifiable genera (Figure 2C). *Sphingomonas* spp., which accounted for 85% of all bacteria was the most predominant bacterial genus in 98% of samples. The second most predominant bacteria were *Nesterenkonia* and *Lactobacillus* with maximum relative abundances of more than 3%. *Acetobacter* (2.9%) was the third most abundant genus, which was predominant in the Ba_1st sample and accounted for more than 85% of bacteria in the sample. Similarly, *Lactobacillus* (more than 65%) was the predominant genus in the Pi1_1st sample. Among the potential opportunistic pathogens, the relative abundance of *Bacillus*, *Clostridium*, *Dietzia*, *Gardnerella*, and *Pseudomonas* was less than 1% (Figure 2D). *Dietzia* and *Bacillus* were detected in 25% and 17% of samples, respectively. *Clostridium*, *Gardnerella*, and *Pseudomonas* were detected in less than 1% of samples. Additionally, *Actinomyces, Agrobacterium, Chryseobacterium, Clostridium, Dietzia, Methylobacterium*, and *Natromonsas* were only present in pineapple samples containing starter culture (LAB and AAB) whereas, these bacteria were absent in non-starter culture containing pineapple samples. Contrarily, *Achromobacter, Anaerobacillus, Delftia, Kocuria, Ochrobactrum*, and *Staphlococcus* were only present in non-starter culture containing pineapple samples and these bacteria were absent in starter culture containing pineapple samples. Two genera including *Lactobacillus* and *Nesterenkonia* showed higher abundance in starter culture containing pineapple samples whereas, *Sphingomonas* abundance was higher in non-starter containing pineapple samples.

Both OTU and ASV taxonomic classifications generated a total of 13 opportunistic pathogenic genera associated with the fermentation processes and products, as shown in Table 2. Among these pathogenic genera, the ASV-based clustering approach identified *Gardnerella* and *Pseudomonas*, whereas the OTU-based clustering approach identified *Acinetobacter*, *Klebsiella*, *Mycobacterium*, and *Streptococcus*. All these bacteria were detected only in the samples from the first stage of fermentation, except *Acinetobacter,* which was detected in the samples from the first stage of fermentation and the final fermentation product. Other pathogenic genera commonly identified in both clustering approaches included *Bacillus*, *Staphylococcus*, *Clostridium*, *Ochrobactrum*, *Cryseobacterium*, *Dietzia*, and *Stenophomonas,* which were detected only in the samples from the first and second fermentation stages, whereas, *Bacillus* and *Clostridium* were also detected from the final fermentation products.

### 3.3. Change in Bacterial Community Composition Concerning Fermentation Processes and Products

The changes in bacterial diversity and abundance at the genus level in the samples from different fermentation stages and the final fermentation products were examined using STAMP software. The analysis revealed that the abundance of *Dietzia*, *Devosia*, *Lactobacillus*, *Nesterenkonia*, *Paracoccus*, and *Pelomonas* was significantly different between the first and second stages of fermentation with the abundance of most genera enriched in the first stage of fermentation (Figure 3). Among these genera, *Lactobacillus* and *Nesterenkonia* were the predominant genera in the first and second stages of fermentation, respectively. *Dietzia*, *Devosia*, *Lactobacillus*, *Nesterenkonia*, *Paracoccus*, *Pelomonas*, and *Stenotrophomonas* were significantly enriched in the first stage of fermentation and the final product. However, the mean proportion of these genera markedly decreased in the final product. Moreover, the comparative analysis of bacterial composition revealed that the abundance of only one genus (*Nesterenkonia*) was significantly different between the second stage of fermentation and the final product. This genus was predominant in the second stage of fermentation, whereas the mean proportion of this genus decreased in the final product.

### 3.4. Prediction of Functional Pathways Based on 16S rRNA Gene Metagenomic Data in Fermentation Stages and Final Products

Functional prediction analysis revealed 5322 KOs, which were grouped into 155 KEGG pathways (level 3). These data were further processed to filter the low variant and extremely low abundant pathways (<4 abundance) using MicrobiomAnalyst. In total, 125 high-quality level 3 KEGG pathways were obtained. The level 2 KEGG pathways were presented in the form of a bubble plot for clear visualization and interpretation (Figure 4). These pathways are mainly associated with six major functional categories (KEGG level 1), namely cellular processes (4.8%), environmental information processing (3.2%), genetic information processing (9.6%), human disease (0.8%), metabolism (78.4%), and organismal systems (3.2%). Additionally, the level 2 KEGG pathways related to metabolism, such as carbohydrate, amino acid, cofactor, and vitamin metabolic pathways, were the significant predominant functions in each sample with a minimum relative abundance of 13%. Among the genetic information processing pathways, replication and repair pathway were significantly predominant with a minimum relative abundance of 11.5% in all fermentation samples. Only one crucial human disease pathway associated with *Staphylococcus* infection was detected in all samples with a maximum relative abundance of 0.6%.

The STAMP software was used to further investigate changes in the metabolic functions, especially during the fermentation stages and in the final product (Figure 5). The comparative analysis revealed that six predicted metabolic functions (environmental adaptation, infectious disease, endocrine system, metabolism of other amino acids, and glycan biosynthesis and metabolism) were significantly enriched (*p* < 0.05) in the samples from the first and second stages of fermentation. However, environmental adaptation and infectious disease were significantly abundant in both stages of fermentation. Moreover, the mean proportion of all significant functions decreased in the second stage of fermentation. Glycan biosynthesis and metabolism, translation, replication and repair, nucleotide metabolism, environmental adaptation, cell growth and death, and folding, sorting, and degradation were significantly enriched between the first stage of fermentation and the final product.

Similarly, the mean proportion of all enriched pathways decreased at the end of fermentation. Additionally, metabolism of terpenoids and polyketides, amino acid metabolism, and membrane transport were significantly enriched in the second stage and final products of fermentation. Furthermore, the abundance of these pathways in the final product of fermentation was lower than that in the second stage of fermentation.

### 3.5. Correlation between the Bacterial Fermentation Community and Predicted Functional Profiles Based on 16S rRNA Amplicon

Pearson correlation analysis was performed to further explore the correlation between the bacterial community and predicted functions and to identify the significant difference between bacterial taxa at the genus level and the KO at level 2. The differences were considered significant and highly significant at *p* < 0.05 and < 0.01, respectively (Figure 6).

The correlation analysis revealed that among all the observed bacterial taxa, *Sphingomonas* was the only predominant genus that was significantly correlated (*p* < 0.01) with all the 25 predicted functions related to fermentation processes and products. Additionally, *Paracoccus* and *Pelomonas* were significantly and positively correlated (*p* < 0.05) with only the endocrine system. In contrast, most bacterial taxa, except *Acetobacter*, were negatively and non-significantly correlated with all predicted functions. *Acetobacter* was significantly and negatively correlated (*p* < 0.05) with the endocrine system. Similarly, *Gardnerella* and *Kocuria* were significantly and positively correlated (*p* < 0.05) with all predicted functions, except cell mobility, transport and co-metabolism, and infectious disease. Additionally, *Deleftia*, which was the fourth most predominant bacterial taxa, was significantly and positively correlated with more than 50% of the metabolic pathways (*p* < 0.01). In contrast, only *Lactobacillus* was significantly and negatively correlated (*p* < 0.05) with cell mobility, cellular community-prokaryote, signal transduction, and digestive system pathways.

None of the identified genera, except *Sphingomonas*, were significantly and positively correlated with the predicted pathway related to infectious disease, which was the focus of this study. However, *Staphylococcus*, a potential pathogen, was not significantly and positively correlated with the pathways related to infectious diseases. *Staphylococcus* was significantly and positively correlated with the endocrine system.

### 3.6. Confirmation/Validation of Potential Pathogens in Fermentation Stages and Products Using PCR

The 16S rRNA amplicon sequencing using ASV and OTU based taxonomic classifications revealed 13 genera belonging to potential opportunistic pathogens associated with the fermentation stages and the final product. To confirm their presence or absence in the fermentation stages and product we targeted the following 13 bacterial species using PCR: *Acinetobacter baumannii*, *Bacillus cereus sensu lato s.l*, *Clostridium difficile*, *Chryseobacterium indologenes*, *Dietzia maris*, *Gardnerella vaginalis*, *Klebsiella pneumoniae*, *Pseudomonas aeruginosa*, *nontuberculous mycobacteria*, *Ochrobactrum anthropi*, *Staphylococcus aureus*, *Streptococcus pneumoniae*, and *Stenotrophomonas*. The PCR analysis revealed absence of all these pathogens except *S. aureus* was detected only in the Pi4_1st sample, which was from the first fermentation stage. The genus *Staphylococcus* was also detected in Pi4_1st sample in 16S amplicon sequencing analysis.

## 4. Discussion

Globally, fermented foods are consumed as ready-to-eat foods. However, the lack of microbial characterization of these food products has raised consumer concerns for food safety, which has led to decreased consumption of these essential food items [50]. Modern high-throughput sequencing identifies only the functional and beneficial microbial diversity of fermented products. Therefore, this study investigated the bacterial diversity and identified potential pathogens in different steps of fermentation and their products using high-throughput sequencing approaches and powerful modern bioinformatics. For improved representation of bacterial composition containing opportunistic pathogens, 16S amplicon sequencing analysis was performed using OTU (USEARCH)-based and ASV (Qiime2)-based taxonomic classification approaches. The OTU-based clustering approach (cut-off value of 97%) yielded 53 classifiable genera from 785,339 sequence reads, whereas ASV-based clustering (cut-off value of 97) yielded 43 classifiable genera from 193,323 sequence reads. The number of genera identified using ASV-based clustering in this study was consistent with the results of a previous study, which reported 43 genera in Jiaosu (fermented fruit and vegetable juice) using OTU-based clustering UCLUST (cut-off value of 98.6%) [51]. The differences in the classification of bacterial genera between these two clustering approaches can be attributed to an increased resolution of ASV-based clustering to rectify the potential sequence errors. In contrast, minimal filtration leads to low-quality sequence reads in the OTU-based clustering approach, which results in a false representation of taxonomic units. Previous studies using different denoising tools and approaches reported a higher number of OTUs than those using ASV-based approaches [52,53,54]. This can explain the genera missed in the ASV-based clustering approach.

Among the genera detected in both clustering approaches, *Sphingomonas* was the predominant bacterial genus in almost 90% of samples and accounted for 80–85% of all identified bacteria. Other studies have reported that *Sphingomonas* is associated with pulque fermentation (80%) and that this was the predominant genus in fermented products of fruit (78%) among all identified genera. *Sphingomonas*, which is considered to be the indigenous flora of soil and plants [55] and is mainly present on the surface of fruits and vegetables, confers resistance and promotes plant growth [53]. A recent study on the fermentation of pickles based on vegetables dehydrated with smoke demonstrated that *Sphingomonas* was the major bacterial genus in the final product [4]. Additionally, the two taxonomic classification approaches yielded two different genera. *Oribacterium* was the second most predominant genus in the OTU-based clustering, whereas *Nesterenkonia* was the predominant genus in ASV-based clustering. *Oribacterium* and *Nesterenkonia* are frequently associated with the surface of fruits, water, and soil. A previous study also reported that these two approaches yield different taxonomic levels, which explains the difference in the discriminatory power of these approaches [56]. Moreover, among all the identified genera, 13 low abundant genera were potential opportunistic pathogens. Most of these genera are predominant or rare flora of fruits, vegetables, and skin and are associated with environmental contamination [57]. Similarly, other studies have reported the presence of most of these opportunistic pathogens in the starter cultures and in different stages and products of various fermentation processes, such as cereals, sausages, vegetables, cheese, and fermented fish [58,59]. The contamination of these genera mainly depends on the manufacturing conditions, such as the time-temperature profile, water activity, and pH applied for fermentation [13]. Other factors that contribute to contamination from pathogenic genera include microbial deterioration and fermented food safety issues, including insufficient sanitation, starter culture quality, and raw material quality [15]. These findings suggest that various factors must be considered to prevent the risk of opportunistic pathogens.

The change in microbial communities during fermentation must be considered to understand the functionality of beneficial microorganisms and the survival of opportunistic pathogens. The results of this study demonstrated that most differentially abundant genera, such as *Paracoccus, Dietzia, Devosia*, and *Pelomonas* were significantly enriched in the samples from the first stage of fermentation, whereas their abundance significantly decreased in the samples from the second stage of fermentation. Similarly, *Dietzia, Devosia, Lactobacillus, Nesterenkonia, Paracoccus, Pelomonas*, and *Stenotrophomonas* were enriched in the samples from the first stage of fermentation but their abundance significantly decreased in the final product. The increased abundance of these bacteria at the initial stages of fermentation can be attributed to the availability of sufficient nutrients and the suitability of environmental conditions, which promote the proliferation and growth of these bacteria. However, these nutrients are depleted and the culture conditions are unfavorable as the fermentation progresses due to decreased pH caused by organic acid producers, such as *Lactobacillus*. These changes decrease the abundance of other sensitive microorganisms [57]. Additionally, *Lactobacillus* was the most abundant genus in the first stage of fermentation, whereas *Nesterenkonia* was the abundant genus in the second stage of fermentation. *Lactobacillus* and *Nesterenkonia* are beneficial for the fermentation of sugars, which leads to acid production and consequently decreased pH levels. Compared with those at the second stage of fermentation, the production of organic acids was upregulated and the pH level was lower in the final product as shown in Appendix A. *Lactobacillus* is a vital probiotic that produces several types of short-chain fatty acids, which are beneficial to human health [60].

To examine the role of microorganisms during fermentation, PICRUSt2 was used to analyze the 16S rRNA sequencing data. Six major functional categories (KEGG level 1) (metabolism, cellular processes (4.8%), environmental information processing (3.2%), genetic information processing (9.6%), human disease (0.8%), metabolism (78.4%), and organismal systems (3.2%)) were significantly associated with the fermentation process. The upregulated function of metabolism suggested the active involvement of microorganisms during fermentation. These results are consistent with those of previous studies, which reported that metabolism was the predominant pathway among the six main KEGG pathways [61,62]. Among the subcategories (KEGG level 2) of these major pathways, carbohydrate, amino acid, cofactor, and vitamin metabolic pathways were the predominant pathways and constituted 13%, while replication and repair functions constituted 11.5%. These inferred functions play an essential role in energy supply, biosynthesis of cellular components, and improving the flavor, taste, and nutritional value of the fermented products [63].

Additionally, the metabolism of other amino acids and glycan biosynthesis and metabolism were the predominant functions between the first and second stages of fermentation. Most functions were significantly enriched in the first stage of fermentation. The functions related to bacterial growth and proliferation, such as replication and repair were the dominant function in the final product. However, amino acid metabolism and metabolism of terpenoids and polyketides were significantly enriched in the second stage and the final product of fermentation. The mean proportion of all significantly enriched pathways decreased in the following order: first stage > second stage > product of fermentation. These results suggest that the bacterial community has varied metabolic capacity and preferential functions in each stage of the fermentation process depending on the fermentation environment. The decline in the mean proportion of these significantly enriched pathways during the fermentation process might be due to the enzymatic activity of various microorganisms. Alternatively, the decline in the number of potential microorganisms responsible for the production of these predicted functions may also contribute to the downregulation of these functions. A similar pattern of decline in the mean proportion of bacterial community was observed in significant microbial shift analysis with the passage of the fermentation process. However, for further confirmation of the presence of these bacterial metabolites and their potential role, we strongly suggest using metabolomics and transcriptomic analyses.

Bacterial function prediction revealed only one crucial predicted human disease pathway associated with *Staphylococcus* infection (KEGG level 3) was detected in all samples with a maximum relative abundance of 0.6%. This proportion is consistent with the results of PCR analysis but less than that reported in the recent study on Chinese traditional fermented food reports. Approximately 2% of bacterial genes in the fermentation process were associated with the human disease, which indicated pathogenic contamination or enrichment during the fermentation process. However, further categories of KEGG pathways and microbes associated with the human disease were not reported in their study [64]. Moreover, *S. aureus* was only detected in the pineapple sample at the first stage of fermentation in ASV-based taxonomic classification and PCR validation assay.

Pearson correlation analysis was performed to examine the correlation between predicted functions and bacterial community of fermentation. The analysis revealed that *Sphingomonas* was correlated with all the predicted functions as this was the predominant genus in fermentation samples and accounted for 80–85% of all observed genera. Additionally, *Sphingomonas* can metabolize various carbon compounds and toxic compounds that are present in low concentrations of nutrients [65]. Additionally, *Delftia* and *Gardnerella* were significantly and positively correlated (*p* < 0.05) with most of the observed functions. In contrast, *Lactobacillus* was significantly and negatively correlated with cell mobility, cellular community-prokaryotes, signal transduction, and digestive system. *Delftia* and *Gardnerella*, which are soilborne bacteria, are commonly found on the surface of fruits. A previous study has demonstrated that these bacteria can metabolize various carbon sources and are correlated with the fermentation processes [66]. Additionally, a significant positive correlation between genus *Staphylococcus* and infectious disease was not observed. However, this bacterium was significantly and positively correlated with the endocrine system. This may be due to the limited number of sequences (0.6%) representing this particular function associated with infectious disease. A previous study demonstrated that differences in the relative abundance of metagenomic data and PICRUSt inferences might not allow an accurate representation of functional traits [67]. Thus, these differences in the functional prediction must be mitigated by increasing the metagenomic sequencing depth.

In this study, the 16S rRNA amplicon was sequenced by targeting the V3–V4 regions of DNA for bacterial identification. The identification of bacteria at the species level based on NGS of 16S rRNA targeting the specific conserved regions of DNA is not reliable as it can yield false-positive results [68]. However, the application of NGS for bacterial identification is reliable up to the genus level. Hence, NGS is the most commonly used robust approach to study microbial diversity in different environmental niches. In this study, 13 species of representative opportunistic bacterial pathogens among the taxonomically identified genera were validated. However, these genera mostly comprised non-pathogenic species. Several studies have demonstrated that the accuracy of PCR assays using appropriate primers for bacterial identification at the species level [69,70]. The PCR analysis revealed that only *S. aureus* was detected in one sample. This indicated that human pathogenic species are not the predominant species in the fermented samples. In this study, the fermented samples were obtained from fruits associated with soil environments. Soil microbes are not considered to be pathogenic. In the genera *Acinetobacter* and *Bacillus*, *A. calcoaceticus*, *A. johnsonii, B. subtilis*, and *B. megaterium* are commonly detected in plants and soil [71]. *C. difficile* is the major pathogenic species in the genus *Clostridium*. However, more than 250 species have also been reported to be beneficial to human health [72]. Although *S. aureus* was detected in the Pi4_1st sample in the PCR assay, it was not detected in the sample from the second stages and the final product of fermentation. Therefore, this hybrid method could ensure that the pathogen was inactive in the fermentation process and the final fermentation product did not contain pathogenic microorganisms. Additionally, the genus *Staphylococcus* was identified in the fermentation stages and the final product using OTU-based taxonomical classification. However, *Staphylococcus* was only identified in the first stage of fermentation using the ASV-based clustering approach. Hence, we propose that compared with the PCR assay, the ASV-based taxonomical classification analysis is more suitable for pathogen prediction.

## 5. Conclusions

This study revealed the correlation between bacterial community composition and functional characteristics during the two stages of fruit fermentation and the final product using high-throughput sequencing and powerful statistical tools. *Sphingomonas* was predominantly detected during all fermentation stages and in the final product. Additionally, *Lactobacillus* was significantly enriched with a higher abundance in the 1st stage of fermentation, whereas *Nesterenkonia* was significantly enriched with a higher abundance in the 2nd stage of fermentation. Functional prediction revealed carbohydrate metabolism was the most abundant metabolic pathway throughout the fermentation process. Moreover, *Staphylococcus aureus* associated with the KEGG human disease pathway was not significantly correlated with opportunistic bacterial pathogenic genera. To confirm the presence of opportunistic bacterial pathogens, PCR assays were performed which revealed absence of these opportunistic pathogens in different fermentation stages or the final products, except in one sample from the first stage of fermentation. Thus, the hybrid method employed in this study can be used to verify the bacterial safety and functions of microorganisms during fermented food production and avoid random detection of opportunistic bacterial pathogens. This hybrid method could provide an effective and high-resolution determinant of the types of bacteria and their functional profile influencing the properties of fermented foods and ensure that no harmful bacteria are present in the final fermentation products.

## Figures and Tables

**Figure 1 microorganisms-09-01574-f001:**
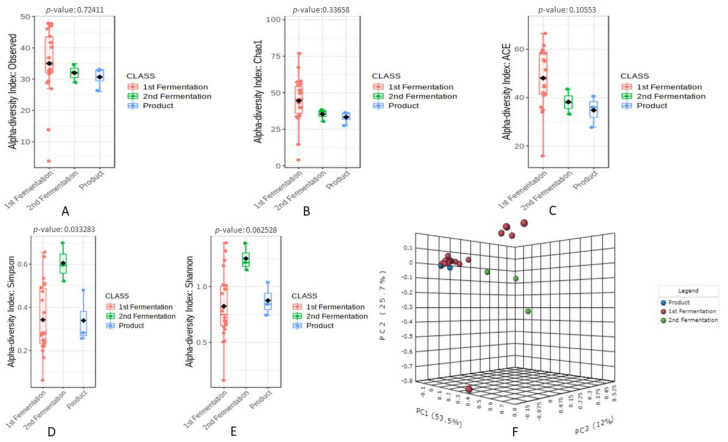
Comparison of bacterial community diversity among the fermentation stages and final product. Alpha diversity was measured by Observed (**A**), Chao1 (**B**), ACE (**C**), Simpson (**D**), and Shannon (**E**) diversity indices whereas, beta diversity was evaluated using PCoA (3D) ordination (**F**).

**Figure 2 microorganisms-09-01574-f002:**
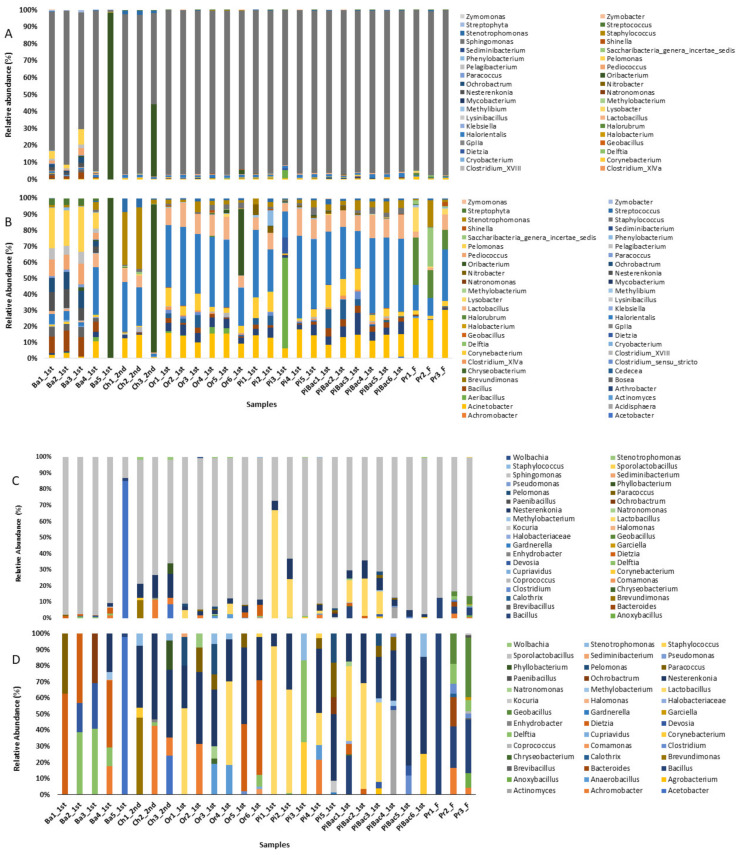
Relative abundance of bacterial diversity at genus level in each fermentation sample using operational taxonomic unit OTU (**A**) and amplicon sequencing variant ASV-based (**C**) taxonomic classifications. The dominant genus *Sphingomonas* was excluded in both OTU (**B**) and ASV-based (**D**) taxonomic classifications.

**Figure 3 microorganisms-09-01574-f003:**
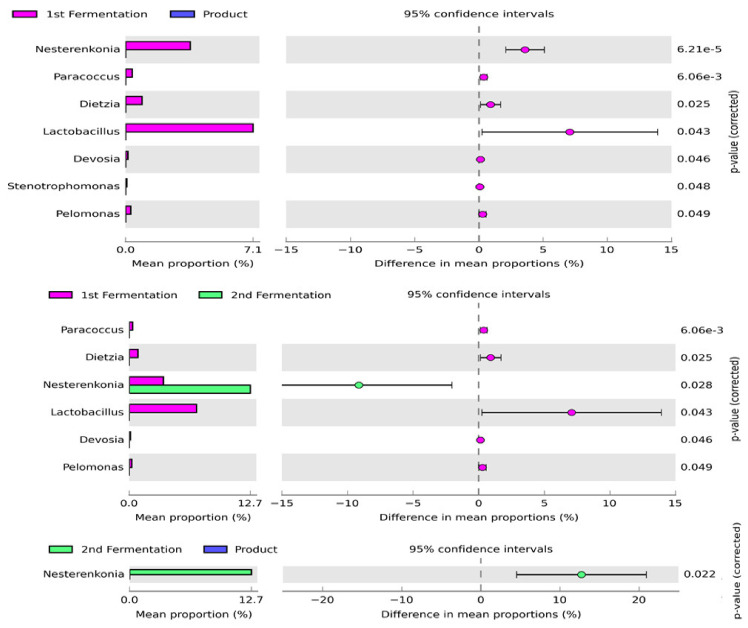
Enriched bacterial genera across the fermentation stages and in the final product. The left panel of these figures shows the abundance of differentially enriched bacterial genera. The right panel represents the significant difference at *p* < 0.05. The middle panel indicates the mean proportion of differentially enriched bacterial genera at a 95% confidence interval.

**Figure 4 microorganisms-09-01574-f004:**
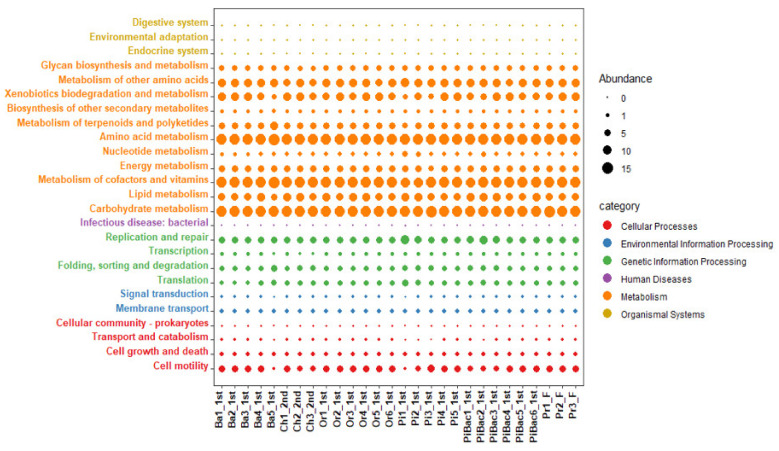
Bubble plot of predicted functions across the fermented fruit samples showing Kyoto Encyclopedia of Genes and Genomes (KEGG) pathway level 2 in different colors on the *Y*-axis belongs to the first category KEGG pathways (legend at right below). The sample names are shown on the *X*-axis. The size and color of the bubble indicate the relative abundance of pathways present in each sample.

**Figure 5 microorganisms-09-01574-f005:**
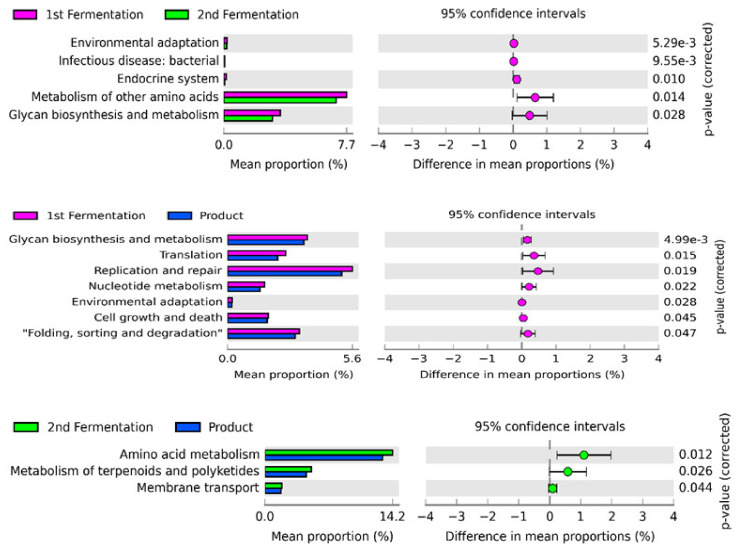
Enriched predicted functions across the fermentation stages and product. The left panel of these figures shows the abundance of differentially predicted functions. The right panel represents the significant difference at *p* < 0.05. The middle panel indicates the mean proportion of differentially predicted functions at a 95% confidence interval.

**Figure 6 microorganisms-09-01574-f006:**
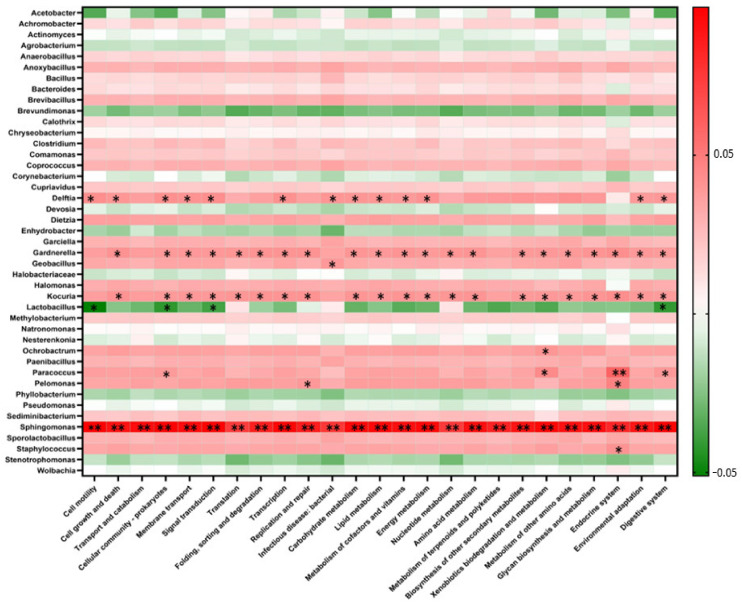
Correlation between the bacterial fermentation community and predicted functional profiles based on 16S rRNA amplicon. Pearson correlation analysis heatmap constructed for each pairwise comparison between level 2 Kyoto Encyclopedia of Genes and Genomes (KEGG) pathways and bacterial taxa at the genus level. The positive and negative correlations are indicated in red and green colors, respectively. The correlation was considered significant at ** *p* < 0.01 and * *p* < 0.05.

**Table 1 microorganisms-09-01574-t001:** Description of samples and fermentation processes based on raw materials and fermentation duration.

Sample ID	Full Name	Process	Fermentation Sampling Time
Or1_1st	Orange-0120	1st stage of fermentation	208 day
Or2_1st	Orange-1272		
Or3_1st	Orange-1266		
PiBac1_1st	Pineapple w/starter culture-0099		67 day
PiBac2_1st	Pineapple w/starter culture-0183		
PiBac3_1st	Pineapple w/starter culture-0151		
Pi1_1st	Pineapple w/o starter culture-0912		60 day
Pi2_1st	Pineapple w/o starter culture-0080		
Ba1_1st	Banana-0174		45 day
Ba2_1st	Banana-0245		
Ba3_1st	Banana-1267		
Or4_1st	Orange-0120		268 day
Or5_1st	Orange-1272		
Or6_1st	Orange-1266		
PiBac4_1st	Pineapple w/starter culture-0099		127 day
PiBac5_1st	Pineapple w/starter culture-0183		
PiBac6_1st	Pineapple w/starter culture-0151		
Pi3_1st	Pineapple w/o starter culture-0242		127 day
Pi4_1st	Pineapple w/o starter culture-0912		
Pi5_1st	Pineapple w/o starter culture-0080		
Ba4_1st	Banana-0174		105 day
Ba5_1st	Banana-1267		
Ch1_2nd	1st stage product No.24	2nd stage of fermentation	240 day
Ch2_2nd	1st stage product No.28		
Ch3_2nd	1st stage product No.30		
Pr1_F	Product No.1	Product	0 day *
Pr2_F	Product No.2		
Pr3_F	Product No.3		

* after completion of fermentation process.

**Table 2 microorganisms-09-01574-t002:** Comparison of pathogenic genera identified using operational taxonomic unit-based and amplicon sequence variant-based clustering approaches and their association with fermentation processes. The symbol ✓ indicates the presence of the genus, whereas × represents the absence.

Genus Names	1st Stage		2nd Stage		Final Product	
	OTU	ASV	OTU	ASV	OTU	ASV
Acinetobacter	✓	×	✓	×	✓	×
Bacillus	✓	✓	✓	×	✓	✓
Staphylococcus	✓	✓	✓	×	✓	×
Dietzia	✓	✓	×	×	×	×
Clostridium	✓	✓	✓	×	✓	✓
Klebsiella	✓	×	✓	×	×	×
Mycobacterium	✓	×	×	×	×	×
Ochrobactrum	✓	✓	×	×	×	×
Chryseobacterium	✓	✓	×	×	✓	×
Stenotrophomonas	✓	✓	✓	✓	✓	×
Streptococcus	✓	×	✓	×	✓	×
Gardnerella	×	✓	×	×	×	×
Pseudomonas	×	✓	×	×	×	×

## Data Availability

In a rigorously anonymous form, raw sequencing data have been deposited in the NCBI depository under the following ID: SUB9829072.

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
