# Peer review of "Deciphering Bacterial Community Structure, Functional Prediction and Food Safety Assessment in Fermented Fruits Using Next-Generation 16S rRNA Amplicon Sequencing"

_microorganisms, 2021, doi:10.3390/microorganisms9081574_

Round 1

Reviewer 1 Report

Summary

In this article, bacterial communities were profiled in fermented fruit over 2 fermentation stages and at the final product. Although bacterial communities have been profiled in other fermented foods such as fermented sausages, there is little existing literature on bacterial communities in fermented fruits. Overall, this study fills that gap. However, there is a lack of clarity on the samples used in this study, which needs improvement.

Broad Comments

  1. More information is needed on the samples and the fermentation process in the methods. Was fruit fermented on its own or was a starter culture or other supplements added? Was the composition uniform between different samples within the same fruit (e.g. Orange-0120 vs Orang-1272)? Do samples from the 2nd stage of fermentation contain all three fruits? Were samples taken longitudinally from 1st to 2nd to final product?

Specific Comments

  1. Introduction: Have human pathogens been found in pineapple, orange, and banana? There should be evidence provided in the Introduction to explain the rationale of the current study.
  2. Line 85: Although molecular approaches can capture viable but non-culturable microbes, they may also capture genetic material from dead cells.
  3. Lines 114-119: The methods state that pH values and sugar and organic acid contents were analyzed, but the results were not reported.
  4. Line 131: 16S rRNA sequence data should be submitted to NCBI.
  5. Line 188: The PCR results for confirmation of the suspected pathogens appear to be missing in the Results section?
  6. Line 196: Italicize Clostridium difficile.
  7. Figure 1 (Line 238): A higher quality image is need. Some of the genus names are difficult to read. “C” and “D” are cutoff.
  8. Figure 2 (Line 272): Was the difference in mean proportions plotted for 2nd Fermentation vs Product?
  9. Figure 3 (Line 299): The figure legend reads that “sample names are shown on the Y-axis”. However in the figure, the sample names appear on the x-axis.
  10. Figure 4 (Line 318): The bottom panel of 1st Fermentation vs 2nd Fermentation appears to be the same that is presented in Figure 2.
  11. Line 354: Delete the word “Authors”.

Author Response

Reviewer’s comments and responses

In this article, bacterial communities were profiled in fermented fruit over 2 fermentation stages and at the final product. Although bacterial communities have been profiled in other fermented foods such as fermented sausages, there is little existing literature on bacterial communities in fermented fruits. Overall, this study fills that gap. However, there is a lack of clarity on the samples used in this study, which needs improvement.

Broad Comments

  1. More information is needed on the samples and the fermentation process in the methods. Was fruit fermented on its own or was a starter culture or other supplements added? Was the composition uniform between different samples within the same fruit (e.g. Orange-0120 vs Orang-1272)? Do samples from the 2nd stage of fermentation contain all three fruits? Were samples taken longitudinally from 1st to 2nd to final product?

Response: thanks for reviewer’s comments. We have made major revision in methodology section according to reviewer's comment (lines: 109-118).

Specific Comments

  1. Introduction: Have human pathogens been found in pineapple, orange, and banana? There should be evidence provided in the Introduction to explain the rationale of the current study.

Response: Thanks for the reviewer’s comment. In the introduction section (line; 58-60), We have already provided evidence that pathogens, such as Listeria monocytogenes, Clostridium spp., Salmonella spp., Staphylococcus spp., Bacillus spp., and Escherichia coli can survive and escape during fermentation and remain infectious. Even there are substantial reports of human pathogens such as E. coli and Salmonella associated with fresh and ready-to-eat pre-cut fruits. However, in the literature, we did not find any evidence of human pathogens in fruit fermentation specifically associated with pineapple, orange, and banana.

  1. Line 85: Although molecular approaches can capture viable but non-culturable microbes, they may also capture genetic material from dead cells.

Response: Thanks for the reviewer’s comments. Yes, it may capture genetic material from dead cells. However, our intended purpose was to differentiate between molecular approaches and traditional approaches to unveil the microbial communities of fermentation processes. Therefore, did not consider the genetic material of dead cells.

  1. Lines 114-119: The methods state that pH values and sugar and organic acid contents were analyzed, but the results were not reported.

Response: Thanks for the reviewer’s comments. We analyzed pH, sugar, and organic acid and alcohol contents from the second stage of fermentation and final products. However, due to the absence of 1st stage of fermentation results, we did not show the data in this manuscript. Now, in the revised manuscript, we have added the results of these parameters obtained from the second stage of fermentation and final products in Supplementary files. Moreover, we had already mentioned in the discussion section (Line; 444-447) that “lower pH and increase acid production was observed in the final product as compared to the 2nd stage of fermentation (data not shown here)”. Now the citation has been changed as we have added the chemical results in the supplementary file to make a possible conclusion based on our chemical analysis.

  1. Line 131: 16S rRNA sequence data should be submitted to NCBI.

Response: Thanks for the reviewer’s comments. We have submitted the DNA sequence data into NCBI database under the following ID: SUB9829072 and this ID has been provided in the Data availability statement in the revised manuscript (lines; 570-571)

  1. Line 188: The PCR results for confirmation of the suspected pathogens appear to be missing in the Results section?

Response: Thanks for the reviewer’s comments. We have added the PCR analysis results in the result section (lines;364-337).

  1. Line 196: Italicize Clostridium difficile.

Response: Thanks for the reviewer’s comment. We have done as per suggested (line: 205.

  1. Figure 1 (Line 238): A higher quality image is need. Some of the genus names are difficult to read. “C” and “D” are cutoff.

Response: Thanks for the reviewer’s comments. We have added higher quality images as per suggested (line;247-248)

  1. Figure 2 (Line 272): Was the difference in mean proportions plotted for 2nd Fermentation vs Product?

Response: Thanks for the reviewer’s comments. We set up the lower limit of difference in mean proportion to visualize the figure that is why higher mean difference did not observe within the set limit. Now we have set higher limit in the revised figures (line: 238) to show all the mean differences.

  1. Figure 3 (Line 299): The figure legend reads that “sample names are shown on the Y-axis”. However, in the figure, the sample names appear on the x-axis.

Response: Thanks for the reviewer’s comments. We have rectified it (lines;307--308)

  1. Figure 4 (Line 318): The bottom panel of 1st Fermentation vs 2nd Fermentation appears to be the same that is presented in Figure 2.

Response: Thanks for the reviewer’s comments. We have changed figure 2 (line;328)

  1. Line 354: Delete the word “Authors”.

Response: Thanks for the reviewer’s comment. We have removed the mentioned word as suggested (line;378)

Reviewer 2 Report

In general, the text is bulky, have many repetitions and lack clear unambiguous descriptions of the data. The aim of the study is not clear. While Lines 94-95 state “This study aimed to investigate the total microbial community and identify the presence of potential pathogens in fermented fruits at different stages of fermentation” and Lines 358-360 state “Therefore, this study investigated the microbial diversity and identified potential pathogens in different steps of fermentation and their products using high-throughput sequencing approaches and powerful modern bioinformatics”, Lines 457-459 state “Moreover, the main aim of this study was to determine the toxic metabolites and potential pathogens in the fermentation stages and products that pose a potential risk to the consumer”. Moreover, throughout the text authors constantly mention the development of some “method” or “hybrid method” or “approach” that can be used for food quality assessments; however, the exact definition of this approach (i.e. order of action, recommendation for interpretation of results) is missing. The data about pH, sugar and organic acid content as well as the results of PCR amplifications, all of which are mentioned in the Materials and Methods section, are not presented anywhere in the Manuscript. Also, there is no clear description of the fermentation process. It is impossible for people who are not employees of “Taiwan Village Enzyme Co., Ltd” to understand metagenome of what exactly was analyzed. Fig.2 and Fig.4 look strange, since some points are outside of the figure’s frame; hence, the sizes of the differences are not seen. The correlation analysis presented in Fig.5 is meaningless in the current situation – there is an extreme overrepresentation of the Sphingomonas. Not surprisingly, their abundance correlates with everything. Additionally, PICRUSt2 only predict possible capabilities of a microbial community. It is incorrect to make conclusions such as in Lines 450-455 and Lines 507-509. You have no information about “enzymatic activity of various microorganisms” or “downregulation of … functions”. You have only “bag of genes” without any further information (transcription, translation, enzymatic activities, metabolites). Also, if you have pathogens and opportunistic pathogens in your fermentation, it is not safe regardless of any bioinformatics exercises. The question is only about “the definition of microbiological limits above which lot rejection is considered” (see for example Pérez-Lavalle, L., Carrasco, E., & Valero, A. (2020). Microbiological criteria: Principles for their establishment and application in food quality and safety. Italian journal of food safety, 9(1).). In conclusion, this article consists of only NGS sequencing data report without any meaningful biology. All conclusions are either blunt data-description or extremely unsupported, speculative statements.

Author Response

Reviewer comments: In general, the text is bulky, have many repetitions and lack clear unambiguous descriptions of the data. The aim of the study is not clear.

Lines 94-95 state “This study aimed to investigate the total microbial community and identify the presence of potential pathogens in fermented fruits at different stages of fermentation” and Lines 358-360 state “Therefore, this study investigated the microbial diversity and identified potential pathogens in different steps of fermentation and their products using high-throughput sequencing approaches and powerful modern bioinformatics”, Lines 457-459 state “Moreover, the main aim of this study was to determine the toxic metabolites and potential pathogens in the fermentation stages and products that pose a potential risk to the consumer”. Moreover, throughout the text authors constantly mention the development of some “method” or “hybrid method” or “approach” that can be used for food quality assessments; however, the exact definition of this approach (i.e. order of action, recommendation for interpretation of results) is missing.

Response: thanks for reviewer’s comments. We have made major revision according to reviewer's comment. We have revised the aims of this study in the introduction section (lines: 94-95, 99-100), and repetitions were removed from the result and discussion sections (lines:293, 470) as suggested. Moreover, throughout the text, now we have considered only the “hybrid method” instead of the "method" or "approach" (lines: 38, 40-42,551). Additionally, we have tried to describe the definition of this hybrid method and its benefits in the abstract (line:38-40). The rest of the things we have tried to describe in the result and discussion sections. More specifically, the order of this hybrid method we have described in this study starts from the microbial characterization of fruit fermentation and their possible functions. Both the microbes and their functions might be beneficial or detrimental to human health. To further confirm predicted microbial functions and their roles, we have recommended more specific approaches such as metabolomics and meta-transcriptomics. To confirm the presence or absence of opportunistic pathogens detected through 16S rRNA, we used PCR as a confirmative test using specific primers.

Reviewer’s comments: The data about pH, sugar and organic acid content as well as the results of PCR amplifications, all of which are mentioned in the Materials and Methods section, are not presented anywhere in the Manuscript. Also, there is no clear description of the fermentation process.

Response: Thanks for the reviewer’s comments. We analyzed pH, sugar, and organic acid and alcohol contents from the second stage of fermentation and final products. However, due to the absence of 1st stage of fermentation results, we did not show the data in this manuscript. Now, in the revised manuscript, we have added the results of these parameters obtained from the second stage of fermentation and final products in Supplementary files. Moreover, we had already mentioned in the discussion section (Line; 444-447) that “lower pH and increase acid production was observed in the final product as compared to the 2nd stage of fermentation (data not shown here)”. Now the citation has been changed as we have added the chemical results in the supplementary file to make a possible conclusion based on our chemical analysis. Additionally, we have added the PCR analysis results in the result section (lines:364-375).

Reviewer’s comments: It is impossible for people who are not employees of “Taiwan Village Enzyme Co., Ltd” to understand metagenome of what exactly was analyzed. Fig.2 and Fig.4 look strange, since some points are outside of the figure’s frame; hence, the sizes of the differences are not seen.

Response: Thanks for the reviewer’s comments. To our best knowledge, it is the first report where we analyzed the diversity of microorganisms involved in fruit fermentation at different stages and the final product using the most widely used 16S metagenomic taxonomic classification approaches such as OTU and ASV. The ASV-based approach was more appropriate in the classification of microbes as we confirmed it using PCR analysis. Additionally, we also predicted possible microbial functions and microbial food safety analysis associated with the different stages and final product of fruit fermentation using 16S metagenomics. Finally, predicted possible opportunistic pathogens through 16S rRNA were confirmed for the presence or absence of these pathogens using PCR assays. Additionally, we have changed fig.2 (line: 283) and fig.4 (line: 328) in the result section. The mean difference pointing outside the frame was due to the lower limits we set during the visualization process in the statistical analysis tool. In the revised figures, we have set a higher limit to show all the mean differences.

Reviewer’s comments: The correlation analysis presented in Fig.5 is meaningless in the current situation – there is an extreme overrepresentation of the Sphingomonas. Not surprisingly, their abundance correlates with everything. Additionally, PICRUSt2 only predict possible capabilities of a microbial community. It is incorrect to make conclusions such as in Lines 450-455 and Lines 507-509.

Response: Thanks for the reviewer comments. Throughout the NGS analysis using both ASV and OTU taxonomic classification, Sphingomonas was in higher abundance than other bacteria. Therefore, it is evident that this bacterium will be overestimated. Nevertheless, the primary purpose of this correlation analysis was to check the contribution of taxa in infectious disease and beneficial to fermentation as we have mentioned Lactobacillus and discussed the pathogenic bacteria such as S. aureus in the description. Additionally, similar approaches of correlation based on Spearman and Person analyses are widely considered in the scientific world to correlate microbial functional prediction with respect to their taxa.

Similarly, PICRUSt analysis has been accepted by the majority of scientists and researchers to predict microbial functions using the KEGG database. However, the difference might be observed because of sequence depth which we have already mentioned in the discussion section (lines: 507-509). Therefore, we have suggested using metabolomics and transcriptomic to confirm these microbial metabolites (lines: 479-481).

Reviewer’s comments: Also, if you have pathogens and opportunistic pathogens in your fermentation, it is not safe regardless of any bioinformatics exercises. The question is only about “the definition of microbiological limits above which lot rejection is considered” (see for example Pérez-Lavalle, L., Carrasco, E., & Valero, A. (2020). Microbiological criteria: Principles for their establishment and application in food quality and safety. Italian journal of food safety9(1).

In conclusion, this article consists of only NGS sequencing data report without any meaningful biology. All conclusions are either blunt data-description or extremely unsupported, speculative statements.

Response: Thanks for the reviewer’s comments. This study was conducted to increase consumer acceptance of fermented fruit foods that are safe from pathogens. In this study, after PCR confirmation, only one sample (Pi4_1st) belonging to 1st stage of fermentation showed S. aureus positive PCR, but not in the 2nd stage and final products (lines: 372-3374). Similarly, S. aureus was only predicted in the same sample using ASV taxonomic classification of 16S metagenomic analysis (lines: 374-375). Therefore, we can say that the final products are free from pathogens as none of the tested pathogens based on 16S rRNA was not confirmed in the final check by PCR (lines: 366-375). For reference, the research article you mentioned dealt with the microbial limits based on CFU in the final product using statistical analysis. We also agree with this kind of approach in assessing microbial safety. However, the NGS-based approach is another way to deal with the microbial safety accepted currently in the scientific world. The descriptions and conclusions we drown in this manuscript are based on the two taxonomic classifications used in NGS analysis (lines: 528-537), Functional prediction using the KEGG database (Lines: 450-462, 481-491) partial chemical analysis (second stage of fermentation and product) (Lines: 434-447), statistical analysis (enrichment analysis of fermentation stages and product based on p= 0.05-0.01; figures, 2, 4, 5) and confirmation of opportunistic pathogens using PCR assays (lines: 364-376, 528-537). 

Reviewer 3 Report

The manuscript propose a hybrid method to be used to evaluate the bacterial diversity, their functional predictions and safety assessment of novel fermentation products. Despite the topic is very interesting as well as the described method, the manuscript has some contradictory parts that needs to be clarified. In my opinion, the manuscript needs to be improved to be accepted for publication.

Line 70: “The recent advances  in next-generation sequencing (NGS) have revolutionized the traceability and characterization of a broad range of pathogens in various fields, including epidemiology, diagnostics, food quality, and forensics” Please add a reference.

Line 73 and line 77: Bacterial nomenclatures should be written in italic. Please carefully check throughout the manuscript

Line 92: fruits and vegetables

Line 113-114: The authors stated that “In the 1st stage of fermentation, these fruits are washed and cut in fermentation tanks with the addition of sugar, yeast, lactic acid bacteria (LAB), and acetic acid bacteria (AAB) following mixing vigorously for 24 hours” but no information were given about the species/strains composition of microbes added, microbial concentration added etc…

Line 115: “pineapple samples without bacteria” better write without microbes.

Line 115: It is not clear why orange and banana samples without microbes were not considered in the study. The authors should better explain the rational behind the experimental fermentation plan.

Lines 116 and 119: The authors stated that “these fermentation tanks are kept at room temperature..” Can the authors specify whether the room temperature was under control or measured during time? What they means for room temperature? What was the range? Temperature is one of the main factor affecting fermentation processes, leading to different microbial populations.

Line 117: The authors stated that the fermentations were carried out for at least 1-9 months. Could the authors explain how they decide the different time of fermentation for the 1st stage? Based on what? How they monitor the fermentation process? Did they perform some physicochemical analysis before 2nd stage?

Line 130-136: I suggest to introduce in the main text of this paragraph entitled “Samples and chemical analysis” a brief description of physicochemical analysis shown in Suppl Table 2.

Line 151-162: In order to help the readers I suggest to show primers sequences in a Table, not listed within the text.

Line 207 and line 210 Acinetobacter baturnannii were detected or identified… Pseudomonas ae-208 ruginosa, Stenotrophomonas maltophilia, Gardmerella vaginalis, and Chryseobacterium indolo-209 genes were identified ….

Figure 1 is not clear. The authors show a panel figure (A-D) with two Fig. 1A and two fig. 1B.

Line 232: “Of the 51 genera, 11 were potential opportunistic pathogens” but in the abstract and also in results the authors stated “a total of 13 opportunistic pathogenic genera” (line 270, line 387 etc)

Line 387-388: Need to be rephrased: The sequencing results revealed that 13 genera were potential opportunistic pathogens associated with the fermentation stages and the final product.

Line 388-396: This part is contradictory and needs to be clarified. At the same time the authors stated that the presence of 13 species of potential opportunistic pathogens were confirmed by PCR (lines 388-393) and in lines 393-396 the authors stated that “The PCR analysis revealed absence of all these pathogens except for S. aureus was detected only in the Pi4_1st sample, which was from the first fermentation stage.”   

Line 440: and in different

Line 462 and Line 524: Lactobacillus should be written in italic

Line 469: short-chain fatty acids

Line 507: Staphylococcus should be written in italic

Line 545: delete “that”

Author Response

Respected Reviewer,

Please find the revised manuscript in the attachment.

Thanks,

Round 2

Reviewer 1 Report

The authors have sufficiently addressed my previous comments.

Author Response

Review #1 comments

The authors have sufficiently addressed my previous comments.

Response: Thank you for your affirmation.

Reviewer 2 Report

Since you clarified the aim of the study as: "This study aimed to investigate the total microbial community and their functional prediction along with the identification of potential pathogens in fermented fruits at different stages of fermentation", I will comment on this exact wording.

Comments on the “to investigate the total microbial community”:

1) As you mentioned Line 111, total microbial community in your case contains not only bacteria but yeasts. What can you say about yeasts? Why didn’t you perform ITS metagenomics?

2) How can you claim that you detect all the bacterial genera without presenting a rarefaction curve? Without the rarefaction curve, how can you be sure that your depth of a sequencing is sufficient to cover all biodiversity in the samples?

3) You didn’t present any metrics of biodiversity (i.e. alpha and beta diversity).

4) There are no comparisons with other data on the fermentation of fruits. For example, why your profile is markedly different from those described in [Ma, D., He, Q., Ding, J., Wang, H., Zhang, H., & Kwok, L. Y. (2018). Bacterial microbiota composition of fermented fruit and vegetable juices (jiaosu) analyzed by single-molecule, real-time (SMRT) sequencing. CyTA-Journal of Food16(1), 950-956.]? What can be the reason?

5) Does fermentation without addition of starter culture (i.e. pineapple fermentation) different in terms of bacterial population from those with a starter addition?

7) Why do the samples Ba5_1st and Ch3_2nd extremely different from the others?

8) Your conclusion in Line 543 “Lactobacillus was significantly enriched in both fermentation stages” contradict your Fig.2 (bar chart). In this figure you have mean proportion of Lactobacillus of approximately 7 % at the first stage of fermentation, near 0 % at the second fermentation stage and near 0 % in the final product. Similarly, you claim that “Nesterenkonia was significantly enriched in the first stage of fermentation and the final products”. However, the mean proportion of Nesterenkonia is approximately 5 %, 13 % and 0 % in the first stage, second stage and product, respectevly.

Hence, the objective to describe "total microbial community" was not fulfilled.

Comments on the “identification of potential pathogens”:

1) What can you say about pathogenic yeasts and molds in your beverage? I think that to really “…increase consumer acceptance of fermented fruit foods that are safe from pathogens”, as you mentioned in your response, you should control these microorganisms too.

2) As you probably aware (since you cite this works – Ref 29), pathogenicity is species or even strain specific property. You have only data about levels of genera. Hence, you NGS data are not very useful in this respect. Moreover, you cite by yourself the Ref 53 which are very pessimistic about using 16S rRNA sequencing especially at the genus level of assignment.

3) Of course, you can argue that you use genera detected in NGS to screen for pathogenic species from only these genera with PCR. In this case, why you didn’t not you PCR screen for Sphingomonas paucimobilis – an occasional human pathogen that is representative of the most abundant genera in your samples.

4) If you claim that you “hybrid method”, consisting of 16S rRNA sequencing followed by PICRUSt2 prediction, is a good method for pathogens detection and food safety assessment, please, provide statistical metrics, which was the point of reference [Pérez-Lavalle, L., Carrasco, E., & Valero, A. (2020). Microbiological criteria: Principles for their establishment and application in food quality and safety. Italian journal of food safety, 9(1)] that I gave you. What is the false positive rate, false negative rate and the limits above which lot rejection is considered? In the current setting, if you did not detect pathogen by 16S rRNA sequencing (false negative, which probability you did not provide) you did not test it with PCR. Hence, you do not reject potentially dangerous (again with unknown probability) lot.

Hence, the objectives of “identification of potential pathogens” and to provide some “hybrid method” for it was not fulfilled.

Comments on the “functional prediction”:

1) As you probably know, since it was constantly mentioned by the authors of PICRUSt2, this pipeline was primarily designed to predict functional potential of Human associated microbial communities. So, my question is as follows: What is the average NSTI values of your samples? Is it sufficiently small to consider your predictions at least plausible for your fermentation community?

2) I did not question 16S rRNA sequencing, PICRUSt2 pipeline or correlation analysis by themselves. I just have a considerable doubt about your way of implementation of this tools and conclusions that you draw from these implementations. Let assume, that your correlation analysis is correct. As you can see on your plot (Fig.5), Sphingomonas is highly correlates with “Infectious disease: bacterial”. Hence, as more of these microorganisms you have as more potential pathogenicity of your microbial community will be. And this is the most predominant genera in almost all the samples. Do you have any comments on this?

Hence, the reliability of your functional predictions and follow up speculations are highly questionable.

Comments on the additional data that you have provided in the supplementary:

1) There is no reference for methodology in Lines 125-128. What is this “previous study”?

2) What organic acid (as you put singular noun here, I presume that it is some particular acid) did you measure and how?

3) I presume (even though it should be clearly mentioned in the Materials and Methods section) that the final product is a mixture of liquids from the second stage of fermentation. Please explain, how can you mix something with ethanol concentration of 0-2.4 (the second stage of fermentation) and get something with ethanol concentration of 2.7-2.8 (the final product)? Where did you get an additional ethanol in the product from? The same for unnamed organic acid. By the way, what is "fermentation process of chelation"? It is definitely not a commonly used term. Please, explain its meaning.

Reviewer 3 Report

The manuscript was sufficiently improved.

Round 3

Reviewer 2 Report

The received answers did not change my opinion, since my main points were not satisfactory addressed. The manuscript has irreproducible experimentation and questionable data analysis. The general form of the manuscript: blunt data description followed by imitation of a discussion - is unacceptable for this journal.

Author Response

Respected Reviewer,

Please find the revised manuscript in the attachment.

Thanks
